# Development of Cracked Egg Detection Device Using Electric Discharge Phenomenon

**DOI:** 10.3390/foods13182989

**Published:** 2024-09-20

**Authors:** Sung Yong Joe, Jun Hwi So, Seung Eel Oh, Soojin Jun, Seung Hyun Lee

**Affiliations:** 1Department of Biosystems Machinery Engineering, Chungnam National University, Daejeon 34134, Republic of Korea; dhrmaksdyd@naver.com; 2Department of Smart Agriculture Systems, Chungnam National University, Daejeon 34134, Republic of Korea; sjha24@naver.com; 3Food Safety and Distribution Research Group, Korea Food Research Institute, Wanju 55365, Republic of Korea; dr51@kfri.re.kr; 4Department of Human Nutrition, Food and Animal Sciences, University of Hawaii at Manoa, Honolulu, HI 96822, USA

**Keywords:** cracked egg, electrical discharge, automation, numerical modeling

## Abstract

Eggs are a highly nutritious food; however, those are also fragile and susceptible to cracks, which can lead to bacterial contamination and economic losses. Traditional methods for detecting cracks, particularly in processed eggs, often fall short due to changes in the eggs’ physical properties during processing. This study was aimed at developing a novel device for detecting egg cracks using electric discharge phenomena. The system was designed to apply a high-voltage electric field to the eggs, where sparks were generated at crack locations due to the differences in electrical conductivity between the insulative eggshell and the more conductive inner membrane exposed by the cracks. The detection apparatus consisted of a custom-built high-voltage power supply, flexible electrode pins, and a rotation mechanism to ensure a complete 360-degree inspection of each egg. Numerical simulations were performed to analyze the distribution of the electric field and charge density, confirming the method’s validity. The results demonstrated that this system could efficiently detect cracks in both raw and processed eggs, overcoming the limitations of existing detection technologies. The proposed method offers high precision, reliability, and the potential for broader application in the inspection of various poultry products, representing a significant advancement in food safety and quality control.

## 1. Introduction

Eggs, which are one of the livestock products in high demand around the world, are an excellent complete food that is not only rich in high quality protein but also contains a variety of nutrients such as vitamins, minerals, and essential amino acids [1,2]. Since eggs are fragile and prone to external and internal defects, careful attention is required during the sorting, packaging, and distribution processes [3,4]. The bloody eggs, cracked eggs, and dirty eggs can be considered abnormal eggs [5]. Dirt and dust on the eggshell can be washed and removed; however, the cracked egg that cannot be restored must be discarded to prevent cross-contamination. If a hairline crack (also known as a “micro-crack”) in the eggshell is not detected during the egg sorting process, this crack can be expanded during distribution [6]. As a result, foodborne pathogens such as *Salmonella* can be penetrated through the expanded crack [7,8]. This poses a significant risk of bacterial contamination and can cause economic losses in the poultry industry [9,10]. In the past, skilled egg inspectors strived to sort the eggs with defects. This inefficient and subjective inspection could incur the distribution of abnormal eggs to consumers [11,12].

As an alternative to human inspection, the mechanical, machine vision, and acoustic response technologies have been widely applied to detect the cracks in the eggshell. In a machine vision system, the images of the eggs are captured by computer vision, and the obtained images are analyzed using an imaging process to identify cracks or dirt [13]. In order to improve the detection accuracy of egg cracks, the machine vision systems have been combined with several technologies, such as artificial intelligence and modified pressure [14]. The acoustic response method is a method for detecting the cracks in the eggshell by analyzing the vibration or sound generated by impacting the eggs with small, striking balls [15]. Recently, it has become possible to quickly detect cracks in continuously transported eggs using the acoustic response method combined with the advanced signal processing technology [16]. The majority of global abnormal egg detection manufacturing companies have adopted and applied the acoustic response method [17]. However, the aforementioned detection methods required the complicated processing technologies to recognize the crack in the eggshell and could not detect the processed eggs, such as hardboiled eggs and baked eggs [18].

In particular, the cracks in the processed eggs cannot be detected by current normal egg detection technologies because the physical properties of eggs are significantly changed during thermal processing [19,20]. It can be assumed that the processed egg with a crack in the eggshell has been deemed normal by current egg detection technologies. In this case, the cracks can be expanded by the physical damages occurring in various processes (i.e., physical shock during distribution, moisture movement and stagnation in the egg during storage, and thermal stress by the cooling process). This can significantly reduce the marketability of processed eggs and result in an adverse effect on public health safety. According to Market Data Forecast, Research & Markets, and Spherical Insights, the global processed egg market is expected to experience steady growth in the coming years. The market size is projected to reach approximately $29.45 billion in 2024, with a compound annual growth rate (CAGR) of about 4.8%, potentially reaching around $34.77 billion by 2028. By 2033, the market size is expected to reach approximately $51.5 billion. Therefore, it is essential to develop novel technology that can detect cracks in the processed eggs. 

Since eggshell is mostly composed of calcium carbonate, which can be a good candidate for high-power insulation, several studies have been conducted to develop new polymeric insulation using eggshells [21]. The eggshell containing a tiny pore has high electrical resistivity under the atmospheric condition; however, the other whole egg components, such as the yolk, albumen, and membrane, can be considered dielectrics because their molecules are partially aligned with the applied external electric field [22]. The eggshell membrane exposed by a crack may be more electrically conductive than the eggshell. Therefore, it was hypothesized that the electric discharge would only be observed at the crack in the eggshell when the egg was placed between high electric fields. As shown in Figure 1a, when the sound eggshell was located between the non-uniform high electric field, the electric discharge was observed all over the surface of the eggshell. On the other hand, the electric discharge was observed at only one crack in the eggshell (Figure 1b), and the trace of electrical discharge appeared in the crack. 

The objectives of this study were to (1) evaluate the effect of electric field for detecting the crack in the eggshell, (2) validate the ES model developed for electric discharge on the crack in the applied electric field, and (3) design and fabricate the cracked egg (including cracks in processed egg) detection system using electric field. The developed crack detection system was expected to be practical for detecting cracks in the processed eggs.

## 2. Materials and Methods

### 2.1. Design and Fabrication of Egg Crack Detection System

An experimental system was developed for detecting cracks in eggs. This system consisted of a crack generation device, a detection unit equipped with electrode pins, a high-voltage power supply, a rotation controller, a detection monitoring device, and a PC (personal computer) for data acquisition and system control. Eggshells, primarily composed of calcium carbonate, were found to exhibit high electrical resistance, rendering them nearly insulative [23]. Conversely, the inner membrane of the eggshell, composed of keratin, was determined to form an organic fiber network that, along with the egg liquid, could act as a conductor [24]. When an electrode with a sufficiently high electric field momentarily passed through a crack, an electrical connection was established between the electrodes, resulting in rapid energy discharge and the generation of an electric spark.

To generate a sufficient electric field for the electric field overshoot phenomenon (electrical spark), a high-voltage power supply utilizing an Insulated Gate Bipolar Transistor (IGBT) module was developed and implemented. The system ensured that the egg rotated and passed the detection unit, enabling the inspection of the entire 360-degree surface of the egg. In actual egg production and sorting facilities, conveyor belts dedicated to eggs facilitated simultaneous rotation and movement. The detection platform used in this experiment was designed for batch processing, where eggs continuously rotated at a fixed position while the detection unit, equipped with electrodes, moved forward for inspection. The detection platform consisted of nine lines, each accommodating six eggs. The detection unit was comprised of six parts, each containing six electrode pins.

A custom load cell system was created to generate eggs with cracks of varying positions, directions, and sizes. Cracked eggs were placed among normal eggs on the detection platform, and experiments were conducted by adjusting the ratio of normal to cracked eggs per line. Current change data resulting from electrical discharge at the cracks were collected through the monitoring system. The light generated by the sparks was converted into data, which then triggered an LED bulb located at the top of the detection unit, indicating the position of the cracked eggs.

#### 2.1.1. Electrode Pins

Eggs are not perfect ellipsoids or spheres and vary slightly in size and shape [25]. Previous studies reported that the thickness of medium, large, extra-large, and jumbo eggs was approximately 0.40 ± 0.02 mm with geometric mean diameters of 44.77 ± 0.59 mm, 46.66 ± 0.62 mm, 48.36 ± 0.62 mm, and 49.83 ± 0.83 mm, respectively. Figure 2a illustrates the size difference between commercially available jumbo and medium-grade eggs. The electrodes had to be arranged in such a way that a uniform high electric field could be applied to the entire surface of the eggs, even when eggs were randomly inserted regardless of their grade. Based on size data for different egg grades obtained from previous research, a set of six electrode pins was used, with each pin spaced 1 cm apart. As shown in Figure 2b, 5 to 6 pins make contact with jumbo eggs, while 4 to 5 pins make contact with smaller eggs. 

During the continuous detection process, when the preceding egg passed through the electrode, the electrode lifted and dropped, potentially impacting and occasionally damaging the following egg. Multi-jointed electrodes, as shown in Figure 2c, were found to be more flexible than single-type electrodes, reducing the impact on the eggs and broadening the crack detection range by adhering more widely to the egg surface. To minimize egg damage, the tips of the electrodes, which contact the eggs the most, were rounded and designed to be wider at the contact points. The electrodes were made of stainless steel (SUS 304), which is lightweight, highly conductive, and resistant to corrosion. The electrode pins were connected by donut-shaped aluminum bases and spacers, with 2 mm thick, 40 mm diameter circular acrylic plates installed between the aluminum bases to prevent interference between the electrodes. 

#### 2.1.2. Power Supply

As shown in Figure 3a–d, the power supply system was comprised of a programmable DC power supply (EX300-12, ODA Technologies, Bupyeong, Incheon, Republic of Korea), a custom IGBT module, a high-voltage switching transformer (30 kv, 15 k–70 kHz, Information Unlimited, Mont Vernon, NH, USA), and an arbitrary waveform generator (Agilent 33220A, Keysight Technologies, Colorado Springs, CO, USA). The IGBT module could be operated in a frequency range between 1 Hz and 20 kHz with an on/off duty cycle of 0.2 to 0.8 and was technically feasible to generate up to 3 kW of pulsed AC. Using a high-voltage switching transformer with an operating frequency range above 15 kHz required waveform conversion using high-speed switching devices, but it was found to be suitable for the high-voltage power supply configuration as it could step up the input voltage by approximately 18.5 times.

Eggshells were determined to be almost insulative, requiring high voltage for micro-crack detection using electrical discharge. The voltage supplied by the DC power supply was converted into square pulses up to 20 kHz through the IGBT module and pulse generator. In the experiment, a 15 kHz frequency was used, and the voltage, transformed into a square pulse, was amplified to a maximum of 3000 V through a flyback circuit before being supplied to the electrodes. Moreover, with the spark generated during micro-crack detection, phenomena such as voltage and current changes were observed. Monitoring devices were essential for measuring these data and ensuring proper voltage supply.

All voltage and current data points were monitored and recorded using a differential probe (PR-60, BK Precision, Yorba Linda, CA, USA), broadband current monitor (169820, Pearson Electronics, Palo Alto, CA, USA), oscilloscope (DPO 4034, Tektronix, Beaverton, OR, USA), and data acquisition device (39704A, Agilent, Palo Alto, CA, USA) as shown in Figure 3e–h. The current and voltage meters were connected to the oscilloscope and data logger to measure the voltage and current supplied to the electrodes from the IGBT-based power supply. The data logger collected the measured voltage and current intensity, while the oscilloscope visually displayed the waveform and magnitude of the measured voltage and current.

#### 2.1.3. Rotation System 

The egg rotation system consisted of a step motor (57HBM100-1000, Leadshine, California, CA, USA) and motor drive (HBS57H, Leadshine, California, CA, USA), a linear actuator module (LSM5-NK173117-0808, Guro, Seoul, Republic of Korea) for moving the detection unit back and forth, and a custom egg rotation roller. The egg rotation roller was designed to ideally contact the egg surface, preventing ejection during rotation due to the non-perfect ellipsoid shape of the eggs. Made of rubber to increase friction and reduce slip, the rollers provided consistent rotation. Despite the tilting phenomenon during rotation due to the non-ellipsoid shape, the elasticity of the rubber prevented breakage. Eleven egg rollers were connected to the step motor. Initially, a fan belt was used to ensure consistent rotation by accurately transmitting the motor’s power, but slip occurred depending on the tension adjustment. This was improved by using gears, as shown in Figure 4a, to ensure consistent rotation across the 11 rollers.

The linear actuator module comprised a linear stage combined with a bipolar motor. The detection unit, with electrode pins, was connected to two linear stages, allowing movement back and forth as illustrated in Figure 4b. While the eggs were continuously rotated by the step motor and egg roller, the detection device with electrode pins was passed over the eggs. Although the electrode pins were placed in the fixed position, the continuous rotation of the eggs on the rollers allowed for the inspection of the entire outer surface of the eggs. By adjusting the rotation ratio of the linear actuator module’s motor and the step motor driving the rollers, the eggs rotated four times while the detection unit passed over them.

### 2.2. Egg Sample Preparation

The medium, large, extra-large, and jumbo-sized brown eggs and the processed eggs were employed in this study. Each 100 eggs (total 500 eggs) were purchased from the local market. The purchased samples were stored at a refrigerated temperature of 4 °C for a maximum of 3 days before being used in the experiments. 

An egg crack striking device was designed and fabricated to obtain the uniform crack in eggshell (Figure 5a). When a flat hammerhead stroked an eggshell, which has a rounded shape, there was a high likelihood of inducing a fracturing effect at the point of contact. Therefore, a hammer with a rounded, striking surface was employed in this study to artificially make the cracks on the surface of eggs. The rounded head allows for the precise application of force at a single point when it comes into contact with the curved surface of the eggshell. The egg was located in the holder and a 100 g impact hammer stroked the surface of the egg. The impacting hammer swing angle was able to be controlled. The impacting force corresponding to the kinetic energy that the egg received was calculated using the following equation [26]:(1)$$F=\frac{{m_{i}}^{2}gh_{i}}{m_{e}}$$
where F is the impacting force (N), $m_{p}$ is the mass (kg) of the impacting hammer, g is the acceleration of gravity (9.81 m/s^2^), $h_{i}$ is the drop height (m) of the impacting hammer, and $m_{e}$ is the mass (kg) of the egg.

In addition, the impacting force was measured by using the customized load cell system, which consisted of an Arduino Uno R3, load cell, HX711 module, and Python UDF. The impacting force and time were recorded and graphed as indicated in Figure 5b. When the impacting hammer swing angle was 30°, the eggshell was cracked, and the impacting force and time were 0.374 N and approximately 0.3 s, respectively. Using a striking machine, it was possible to create not only in the latitudinal and longitudinal directions of the eggshell but also the tip of the egg (Figure 5d). 

### 2.3. Mathematical Modeling

The inner membrane of the eggshell, made of keratin, formed an organic fiber network and acted as a conductor along with the egg liquid under normal conditions. Cracked eggs had a short air path connecting the shell to the inner membrane along the crack. For dielectric materials like air, when the electric field strength exceeded a certain value, the binding charges were forced to flow, causing insulation breakdown [27]. To understand this behavior, a simulation based on mathematical modeling was performed. Numerical studies of the model and identification of related equations revealed the fundamental interactions between the electric field, charged particles, and neutral particles.

The numerical modeling of electric field and charge density distribution in COMSOL Multiphysics involved coupling two physical phenomena (electromagnetic and electrostatic models), achieved by solving the following governing equations. The modeling process included creating the geometric model, assigning initial and boundary conditions, generating and optimizing the mesh, selecting the solver, setting tolerance and time steps, and achieving a built-in convergence solution [28,29].
(2)∇·D=ρ
(3)∇·E=0
(4)E=−∇·V
where D is electric displacement (C/m^2^), ρ is space charge density, E is electric field strength (V/m), and V is the electrical potential (V).

In the case of linear materials, E is directly proportional to D, which is presented as:(5)$$D=\varepsilon E=\varepsilon_{0}\varepsilon_{r}E$$
where $\varepsilon_{0}$ is the vacuum dielectric constant ($8.9\times 10^{-12}$, F/m) and $\varepsilon_{r}$ is known as relative permittivity, and it is one of the materials’ properties. But in the case of nonlinear material, this relationship is presented as:(6)$$D=\varepsilon_{0}\varepsilon_{r}E+D_{r}$$

$D_{r}$ is known as remanent displacement, and it is the displacement at the absence of the electric field. In order to find a distinctive solution, it is necessary to consider the boundary conditions as well. The boundary conditions, would represent the interface between different media and follow the following equations:(7)$$n_{2}\cdot (D_{1}-D_{2})=\rho_{s}$$

In which $n_{2}$ is the outward normal from medium two. Different mediums behave differently when it comes to electric charges. In dielectric materials, charges can displace within atoms or molecules, although this displacement is not nowhere near charges’ movements in conductors. By applying an external electric field to a dielectric material, the positive charges of its molecule are displaced along the field and negative charges are displaced in the opposite direction of the field.

### 2.4. Mathematical Modeling Setup

The electromagnetic field and charge density distribution were analyzed in two forms: a static 3D type with an air layer between the electrode and membrane and a dynamic 2D type with the electrode directly contacting the membrane. The boundary conditions of the electromagnetic equations assumed insulation from all external geometries, with an initial temperature T_0_ = 303.15 K and an initial potential V_0_ = 0 V. Stainless steel used for the electrodes had a relative permittivity of 1 and an electrical conductivity of 1.45 × 10^6^ S/m, as per the material library. The relative permittivity and electrical conductivity values for the eggshell and membrane were set based on previous studies. The eggshell had a relative permittivity of 15 and an electrical conductivity of 0.5 S/m, while the membrane had a relative permittivity of 3.6 and an electrical conductivity of 1.5 S/m [30,31]. The air had a relative permittivity of 1 and an electrical conductivity of 0 S/m. During dielectric breakdown, the air layer’s conductivity matched that of the membrane. The computational domain was discretized using tetrahedral mesh to enhance mesh quality. The direct linear system solver (PARDISO) was used to increase convergence rates, with relative and absolute tolerances set to 0.5 each. The simulation geometry is schematically shown in Figure 6.

## 3. Results

### 3.1. Electrical Discharging Detection 

To visually detect sparks more easily, higher voltage and current levels were required. Using higher voltage could facilitate the electrical breakdown of the air layer between the crack and the membrane and was possible to increase the rotation speed and movement of the eggs, thereby enhancing detection speed. However, it also could induce the risk of current fluctuations depending on the crack size, affecting detection accuracy and user safety. Therefore, it was necessary to determine the optimal voltage that could generate visible sparks. It was observed that a voltage exceeding 500 V between the electrode and the crack produced visible sparks. The power supplied to the electrode pins was set to over 500 V at a frequency of 15 kHz. Cracked eggs generated for the experiment were placed among normal eggs and passed through the detection unit. Visible sparks were observed at the cracks. Figure 7a,b show the variations in the supplied voltage and current to normal and cracked eggs. The waveform at the top represents the supplied voltage, while the waveform at the bottom represents the measured current. When 500 V with 15 kHz frequency was applied to normal eggs, the measured average current was 0.65 A; however, in cracked eggs, the current was suddenly increased up to a maximum of 1.5 A. 

As shown in Figure 8b, the sparks generated by the electrical discharge created the traces at the location of the cracks. The traces corresponded to the length or shape of the cracks, making it easy to identify their positions. Evidence of electrical discharge was also observed on the inner membrane of the egg, which was in contact with the cracks. The sparks were presumed to occur as momentary currents flow through the membrane, which has significantly lower electrical resistance than the eggshell. However, the cracks were not wide enough for the electrode pins to make direct contact with the exposed membrane. It was presumed that after dielectric breakdown occurs in the thin air layer between the electrode pins and the membrane, momentary current flows through the membrane, completing the electrical discharge mechanism. This series of processes was able to explain that electrical discharge did not occur below a certain voltage threshold, but sparks were observed at voltages above 500 V. 

According to the research completed by Etuck, S. E., the relative permittivity of the egg membrane decreases as the frequency of the applied voltage increases [32]. At frequencies above 10,000 kHz, the membrane’s permittivity was measured to be similar to that of air. At frequencies above 100,000 kHz, it was observed to be less than the relative permittivity of air, which is 1. Therefore, the high-frequency power supply can reduce the relative permittivity of the membrane exposed through the crack, allowing current to flow more easily through the membrane and facilitating the generation of electrical sparks. This method eliminates the need for a sequential process where electrical breakdown of the air layer is followed by current flow through the membrane to generate sparks. It would be a much faster and safer method for detecting cracked eggs compared to using high voltage. However, with current technology, it is challenging to develop a power supply that can consistently and continuously deliver the desired power while withstanding momentary current changes at frequencies above 100,000 kHz. Therefore, it was necessary to explore methods that could accurately detect the presence of small sparks and optimize the detection process to replace the current method of visual confirmation.

The experiment on the variation in the number of cracked eggs was conducted by placing cracked eggs on all five detection lines, as shown in Figure 9a. Out of 30 cracked eggs, sparks were detected in 29 eggs. For the one egg that was not detected, a separate detection test was conducted, and sparks were successfully detected when the egg was tested individually. The detection units were connected in parallel, and in a parallel structure, the electrical current tends to flow along the path of least resistance, where it could move more easily. Consequently, more current was supplied to eggs with larger cracks, which likely resulted in insufficient current to generate sparks in eggs with relatively smaller cracks. Figure 9b shows the traces of sparks according to the crack width. During the detection process, eggs with wider cracks continuously generated sparks while rotating, whereas eggs with narrower cracks occasionally produced sparks. Accordingly, distinct linear traces were observed in eggs with wide cracks, while eggs with narrower cracks exhibited dot-like traces. The difference in crack size affected the frequency and duration of spark generation. However, regardless of the crack size, the magnitude of the varying current was consistently observed to be a maximum of 1.5 A.

Additional scenarios, such as cooked eggs and washed eggs with residual water, were tested for detection accuracy. Figure 10a shows the internal structure of processed and raw eggs. Although the internal properties of processed eggs change during processing, the position and electrical characteristics of the membrane remain unchanged. Therefore, sparks were generated if there were cracks in the eggshell, regardless of whether the egg was processed or not. However, no electrical discharge was detected in eggs that had been left for an extended period. As shown in Figure 10b, it was assumed that the membrane in these aged eggs dried out, resulting in insufficient electrical conductivity. This phenomenon is related to the membrane’s electrical conductivity and can occur regardless of whether the egg is processed.

Figure 10c shows an egg with external contamination. Since sufficiently dried contaminants could not conduct current, no false detections occurred. However, when compared with other results, it was presumed that electrical discharge may occur due to external contamination if moist contaminants are present on the egg. The majority of sorting processes, including crack inspection, have been conducted after the eggshells have been thoroughly cleaned to remove any debris. Therefore, it was concluded that the impact of false detections caused by moist contaminants is minimal. Figure 10e,f illustrates the voltage and current waveforms for eggs soaked in water. While the waveform fluctuated slightly due to the moisture on the eggshell, no electrical discharge occurred in the absence of cracks, despite the external moisture. Despite concerns that residual moisture on washed eggs might reduce detection accuracy, the sparks occurred precisely at the crack locations. The magnitude of current change during the discharge phenomenon was similar to that of untreated standard samples. It was assumed that residual moisture within the cracks was more likely to positively influence the occurrence of electrical discharge.

### 3.2. Optimizing Flame Observation 

The electric discharge phenomenon involved various combustion reactions, manifested as heat, smoke, combustion gases, and radiation. Heat radiation emitted light, with wavelengths varying based on flame temperature, including ultraviolet, visible, and infrared light [33]. High-resolution optical analyzers (sensors) were used to identify specific light sources corresponding to fire characteristics using these optical properties.

To facilitate spark detection from the detection unit, an IR sensor responsive to 760–1100 nm wavelengths was employed. The flame detection system consisted of an Arduino Uno R3, Flame sensor_C57, and an LED bulb. When the detection unit encountered a crack, the spark generated was detected by the sensor and converted into data. Signals exceeding the threshold were used to trigger the LED bulb’s ON/OFF state. If the IR sensor sensitivity was set too high to detect small sparks, it became susceptible to ambient light and sparks from adjacent electrodes, resulting in false positives (Figure 11a). Conversely, lowering the sensitivity reduced false positives but also decreased the accuracy in detecting small sparks.

To minimize false detections from ambient light and adjacent electrode sparks, a 3D-printed case was designed to enclose the electrodes as shown in Figure 11b, with the sensor fixed 3 cm above the contact point. This case blocked environmental factors and amplified the light from electrical discharge, providing reliable trigger signals.

The crack detection device incorporating the sensor lowered the required voltage for detection. The need for manual detection of electrical sparks by the operator was eliminated, and small sparks, which are difficult to visually confirm, were reliably detected. High detection accuracy was demonstrated regardless of the location and size of the crack in the egg. As the length of a crack increases, the number of electrodes making contact also increases, resulting in more spark reactions, which makes detection easier. Additionally, the larger the crack area, the more continuous the sparks, facilitating easier detection. Therefore, it is essential to focus on detecting smaller cracks, which are more challenging to identify. Although most egg cracks appeared in the broad middle section, some eggs had cracks at the ends. Cracks at the ends were more likely to be missed than those in the middle. Also, as shown in Figure 12c, if the crack is smaller than the 1 cm spacing between the electrode pins, there is a possibility that it may not be detected. These issues are expected to be resolved by increasing the number of electrode pins in the detection unit and reducing the spacing between the pins, which would allow for the detection of smaller cracks.

### 3.3. The Simulated Electric Field Strength Distribution 

According to the fundamental principles of electrical breakdown, increasing the voltage applied to an insulator under a specific electric field rapidly increased the number of charge carriers and decreased resistivity, resulting in strong current generation. The breakdown voltage of an air dielectric was much lower than that of an eggshell, making it likely that a thin air layer between the electrode and membrane would cause electrical breakdown when high voltage was applied to the crack. Upon voltage application, the injected space charge induced an internal electric field [34]. The electric field distribution with an air layer between the electrode and membrane was shown in Figure 13a. The electric field was observed to be 18,000 to 19,000 V/m up to the air layer with similar relative permittivity but did not extend to the membrane with a higher relative permittivity. Figure 13b illustrates the space charge distribution, showing a sharp increase in electric field distortion at the air layer interface. The highest electric field strength, leading to dielectric breakdown, was observed in the air layer. Following dielectric breakdown, as current flowed through the membrane, space charge accumulation was more pronounced near the membrane, as shown in Figure 14b.

The variation of the electric field as the electrode passed through the membrane was simulated in Figure 15a. The low conductivity of the eggshell prevented electric field distortion until the electrodes contacted the membrane, causing distortion along the membrane, particularly at the bends. Figure 15b shows the charge density, with the highest accumulation at the electrode before membrane contact, followed by a decrease as current flowed through the membrane. If a high-frequency power supply could be configured, the characteristics of the air layer could be ignored. Instead, direct contact between the electrode and the membrane could be assumed, as indicated by the simulation results. This approach eliminated the need for the sequential current flow following charge accumulation and dielectric breakdown of the air layer, as illustrated in Figure 13 and Figure 14.

## 4. Discussion

The primary focus of this study was to propose a novel method for detecting cracks in eggshells based on discharge analysis. The method developed in this study offered high precision, stability, and low dependency on environmental conditions. This section will further discuss the electrical properties of poultry eggs and explore the generalization and universality of the proposed method.

The system developed in this study could be directly attached to egg crack detection conveyors, providing greater convenience compared to existing detection devices and systems. At a roller speed of 120 RPM and 1.8 m/min, the eggs contacted the electrodes an average of four times, resulting in spark generation regardless of crack location. This enabled the detection of approximately 6000 eggs per hour. With optimized voltage and advanced detection methods, accurate crack detection could be achieved with only two contacts, significantly increasing processing speed and potentially replacing existing detection devices.

Crack detection technology based on electrical properties held significant research value and market potential for future agricultural product quality inspection. Although this study focused on eggs, the method could be applied to various poultry products, demonstrating its versatility and generalizability [35]. Additionally, the system maintained consistent detection accuracy regardless of the egg’s processing state, overcoming the limitations of current commercial technologies. Cracks in eggs posed a risk of contamination by harmful microorganisms, including *E. coli*. The high heat generated by electrical sparks may aid in sterilization, which warrants further investigation. While the current spark response detection relies on IR sensors to detect wavelengths, future studies will explore using current changes as trigger signals. This method would not be affected by environmental light conditions and is considered optimal for field applications.

## 5. Conclusions

In this study, a discharge-based detection device was developed to quickly and accurately detect cracks in eggs. The electrodes were designed to be flexible to prevent damage to the eggs. An IGBT-based power supply was developed to apply high voltage. Electrical discharge occurred due to the difference in electrical conductivity between the eggshell and the exposed membrane at the cracks. Sparks were generated at the cracks, causing a rapid change in current and leaving marks that indicated the location of the cracks. The discharge method was found to be applicable to all poultry products, regardless of their processing state. Numerical simulations were used to verify the effects of charge on the identification of cracked eggs. These simulations investigated the distribution of the electric field and charge density when electrical breakdown occurred in the air layer, allowing current to flow.

## Figures and Tables

**Figure 1 foods-13-02989-f001:**
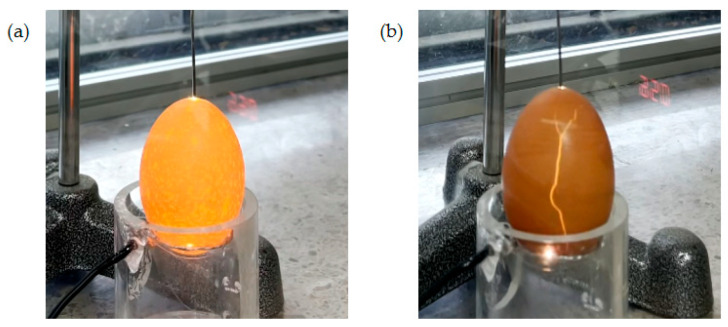
Electric discharge of non-uniform high electric field in the eggshell: (**a**) sound egg; (**b**) cracked egg.

**Figure 2 foods-13-02989-f002:**
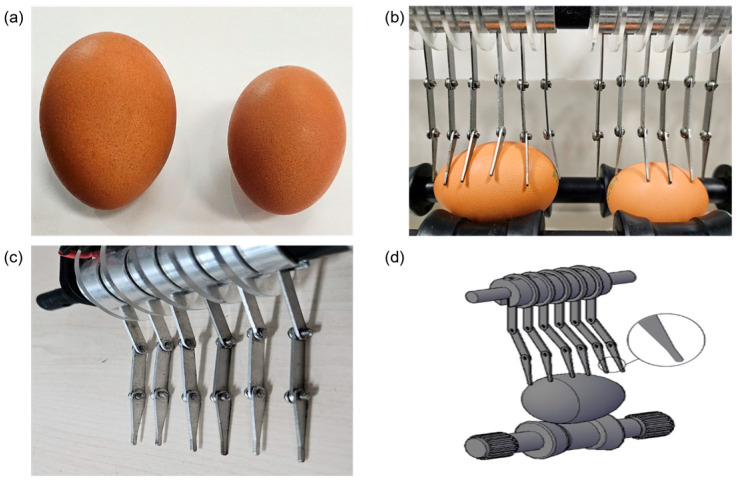
Designed and fabricated electrode pins: (**a**) size differences by egg grade; (**b**) electrode contact appearance according to size difference; (**c**) fabricated electrode shape; (**d**) schematic diagram of electrode body and electrode tip.

**Figure 3 foods-13-02989-f003:**
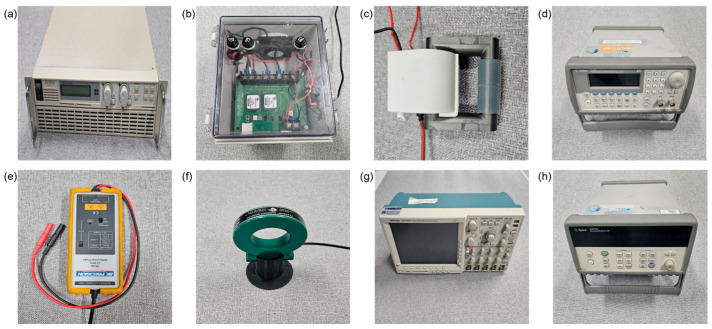
Configuration of power supply and monitoring device: (**a**) programmable DC power supply; (**b**) custom IGBT module; (**c**) high-voltage switching transformer; (**d**) arbitrary waveform generator; (**e**) differential probe; (**f**) broadband current monitor; (**g**) oscilloscope; (**h**) data acquisition device.

**Figure 4 foods-13-02989-f004:**
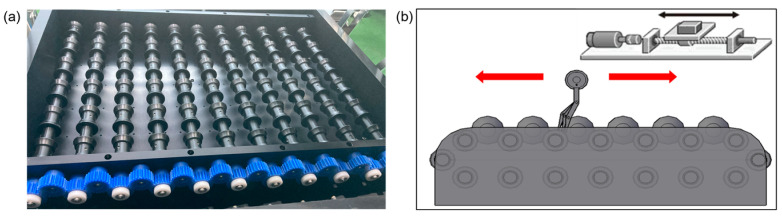
Design and fabrication of rotating systems: (**a**) egg rotating roller and power transmission gear; (**b**) design for moving the detection unit using a linear actuator.

**Figure 5 foods-13-02989-f005:**
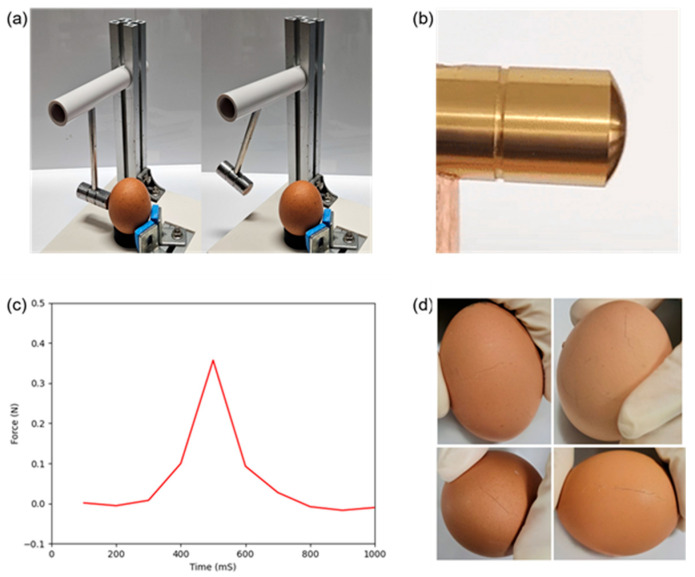
Egg crack generation: (**a**) egg crack striking device; (**b**) hammer shape; (**c**) impacting force; (**d**) types of cracks.

**Figure 6 foods-13-02989-f006:**
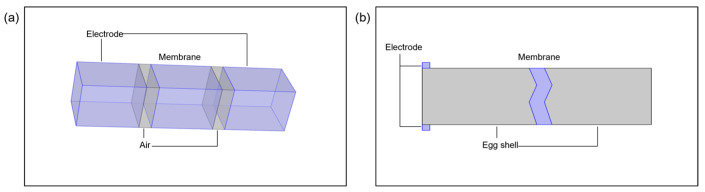
Simulation geometry: (**a**) 3D type; (**b**) 2D type.

**Figure 7 foods-13-02989-f007:**
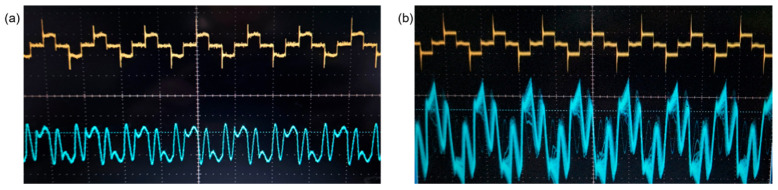
Supplied voltage and current waveforms: (**a**) power waveform for normal eggs; (**b**) current waveform variation in cracked eggs.

**Figure 8 foods-13-02989-f008:**
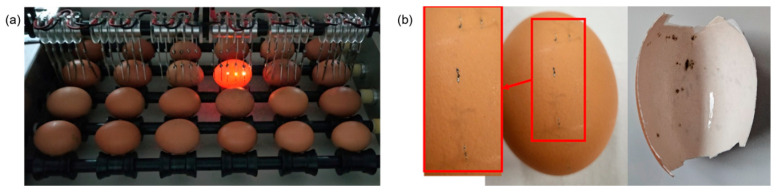
Detection of flame reaction in broken eggs: (**a**) flame reaction caused by electrical discharge; (**b**) traces of electrical sparks left in cracks variation in cracked eggs.

**Figure 9 foods-13-02989-f009:**
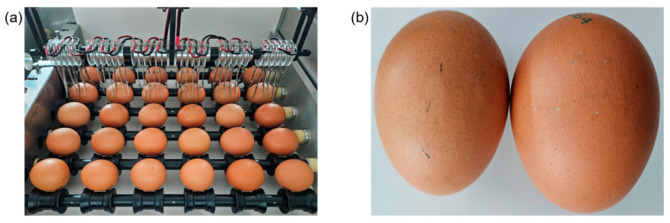
Detection of cracked egg: (**a**) 30 broken eggs layout; (**b**) flame traces according to crack size.

**Figure 10 foods-13-02989-f010:**
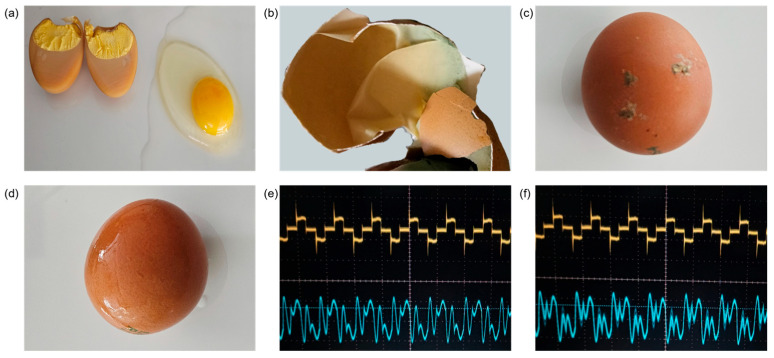
Additional experimental scenarios: (**a**) changes in the properties of the material inside the egg; (**b**) dried egg membrane; (**c**) contaminated egg; (**d**) water-soaked egg; (**e**) standard power waveform; (**f**) waveform rippled by moisture.

**Figure 11 foods-13-02989-f011:**
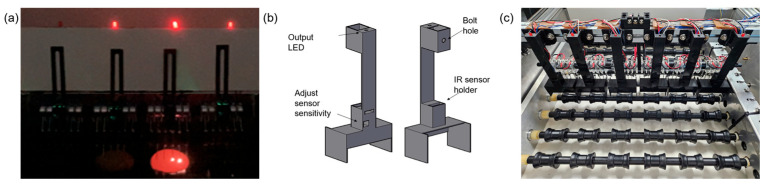
Flame reaction detection automation design: (**a**) misreaction due to surrounding environment; (**b**) case design; (**c**) detection device with case installed.

**Figure 12 foods-13-02989-f012:**
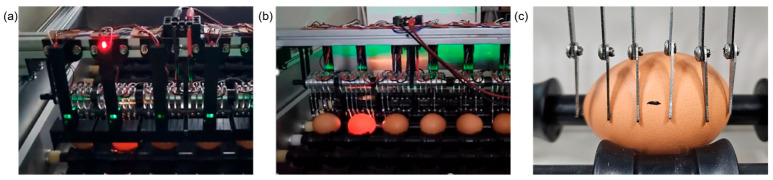
Electric spark reaction detection experiment: (**a**) the view from the front; (**b**) the view from behind; (**c**) micro-cracks.

**Figure 13 foods-13-02989-f013:**
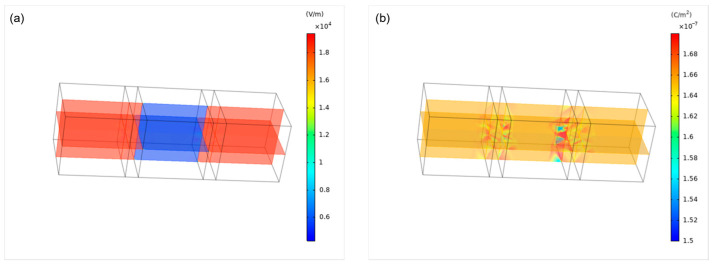
Electric field simulation before dielectric breakdown: (**a**) electric field distribution; (**b**) charge density.

**Figure 14 foods-13-02989-f014:**
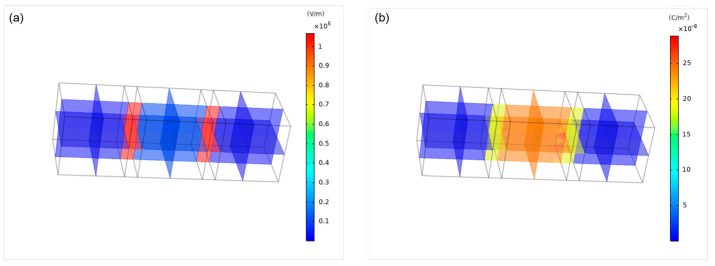
Electric field simulation after dielectric breakdown: (**a**) electric field distribution; (**b**) charge density.

**Figure 15 foods-13-02989-f015:**
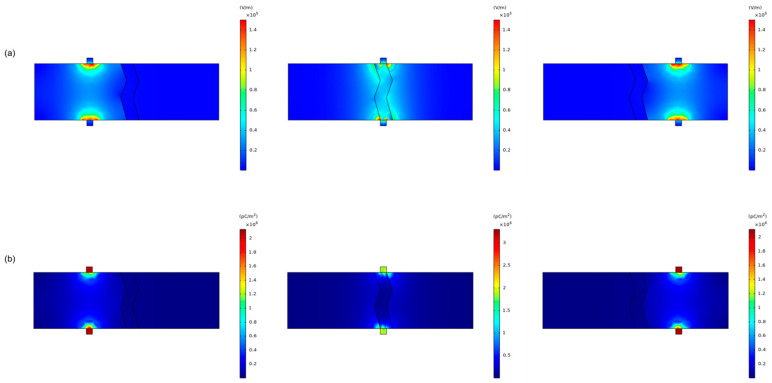
Electric field simulation: (**a**) electric field distribution; (**b**) charge density.

## Data Availability

The original contributions presented in the study are included in the article, further inquiries can be directed to the corresponding author.

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
