# Peer review of "Development of Cracked Egg Detection Device Using Electric Discharge Phenomenon"

_foods, 2024, doi:10.3390/foods13182989_

Round 1
Reviewer 1 Report
Comments and Suggestions for Authors
This study was to propose a discharge analysis method for detecting cracks in eggshells and demonstrates the use of discharge -based device for crack egg detection. There are matured acoustic or vision-based methods, but authors overemphasize the shortcomings of these detection techniques. This was not a good analytical attitude in the section Introduction and Discussion. Is this method effective for detecting crack eggs? Here needs some supporting evidence.
1 In figure 2, the length or rotation angle of egg on the roller was contacted by electrode pins ? Here need to classify ? When the crack was not contacted, can it be detected ?
2 Please provide the electronic signal for sound and crack eggs ?
3 Please provide the relationship between the length or area of egg crack and the electrical signal ? or the relationship between crack grade and the electrical signal .
4 Please Provide the test data (accuracy and confusion matrix) for supporting the effective of discharge method ?
5 If the surface of egg has bumps caused by stains or increased conductivity, will this accurately detect whether the egg is playing well ? Please provide the test data.
Author Response
The authors appreciate the reviewer’s constructive comments and suggestions. The revised sentence was written with a different color in the revised manuscript.

Reviewer 2 Report
Comments and Suggestions for Authors
This paper describes an original technique for detecting cracks on eggs. The principle is to use the difference in conduction between the shell and the inside of the egg. The major problem is to apply the potential to the egg to be tested, which is ovoid. The problem is well described and explained. However, the part relating to the electrodes, an essential element of the device, could be more explicit. A calculation code was developed with COMSOL. This part does not provide any new information, especially since it does not take into account the very particular geometry of the egg.
Author Response

(The authors gave the same response as above.)

Reviewer 3 Report
Comments and Suggestions for Authors
The manuscript foods-3148922 deals with the development of cracked-egg detection device using electric discharge phenomenon. Although the topic is interesting with potential commercial implications, my recommendation is that major revision is needed. I suggest that authors checkup a few things as follows.
· L16: ‘detective eggs’ it may be ‘defective eggs’, please check it.
· Please write abstract in the following order: background, objective(s), materials and methods, and results and discussion. The current abstract is unclear to the reader.
· L39: ‘eggs with dirty” may rewrite as “dirty eggs”
· L39-40: “Dirty and dust on the eggshell” may change to “Dirt and dust on the eggshell”
· L51: Change “to identify the cracks or the dirty” to “to identify cracks or dirt”
· L71-74: Please update with the latest data, particularly “the global processed egg market size was estimated at approximately 17 million tons in 2015”.
· Figure 1 should be moved from the Introduction section to the Results and Discussion section. If the figure is not from the authors work, the source should be credited in the manuscript and figure may remain in the Introduction section.
· L103: please provide the full form of “PC”
· L215-216: “uniform crack in eggshell (Fig. 2a)” it must be Fig. 5a, please check it and make the necessary changes.
· Fig. 5a: Is the surface of the striking hammer flat? How do you ensure that the impact from the striking hammer produces a crack line in the eggshell? It is possible that there may be a crushing effect at the point of contact between the hammer and the eggshell without creating a significant crack line.
· L253: “electric displacement (C/m2)” write the unit as “C/m2”
· I strongly recommend including a schematic drawing or picture of the whole system setup to demonstrate its working principle.
· Did authors control the size of crack in the eggshell? What is the relationship between the size of crack and the electric discharge?
Comments on the Quality of English LanguageEnglish language can be improved
Author Response

(The authors gave the same response as above.)

Round 2
Reviewer 1 Report
Comments and Suggestions for Authors
In this revised manuscript, I did not found any improvement work had been made to provide experimental data to increase the feasibility of practical operation of this method.
In addition, the author also failed to answer the questions raised by the reviewer correctly and did not answer the questions asked. The manuscript provides a simulated layout diagram, but it is not a measured layout diagram. There should be a careful comparison and analysis.
If there is short of experimental data or results provided, this manuscript would be rejected in next round reivew.
Author Response
The authors appreciate the reviewer's constructive comments and suggestion.

Reviewer 3 Report
Comments and Suggestions for Authors
The manuscript is well revised
Author Response

(The authors gave the same response as above.)
